# Women's childbirth experiences in the Swedish Post-term Induction Study (SWEPIS): a multicentre, randomised, controlled trial

Helena Nilvér ![] ,[1] Anna Wessberg,[1,2] Anna Dencker,[1] Henrik Hagberg,[2,3] Ulla-Britt Wennerholm,[2,3] Helena Fadl ![] ,[4] Jan Wesström,[5] Verena Sengpiel ![] ,[2,3] Ingela Lundgren,[1,2] Christina Bergh,[6] Anna-Karin Wikström,[7] Sissel Saltvedt,[8,9] Helen Elden ![] [1,2]

► Prepublication history and additional materials for this paper are available online. To view these files, please visit the journal online (http://dx.doi. org/10.1136/bmjopen-2020- 042340).

For numbered affiliations see end of article.

**Correspondence to**
Helena Nilvér;
helena.nilver@gu.se

## ABSTRACT

**Objective** To compare childbirth experiences in women randomly assigned to either induction of labour at 41 weeks or to expectant management until 42 weeks, in the Swedish Post-term Induction Study.

**Design** A register-based, multicentre, randomised, controlled, superiority trial.

**Setting** Women were recruited at 14 hospitals in Sweden, 2016–2018.

**Participants** Women with an uncomplicated singleton pregnancy were recruited at 41 gestational weeks.

**Interventions** The women were randomly assigned to induction of labour at 41 weeks (induction group, n=1381) or expectant management until 42 weeks (expectant management group, n=1379).

**Outcome measures** As main outcome, women's childbirth experiences were measured using the Childbirth Experience Questionnaire version 2 (CEQ2), in 656 women, 3 months after the birth at three hospitals. As exploratory outcome, overall childbirth experience was measured in 1457 women using a Visual Analogue Scale (VAS 1–10) within 3 days after delivery at the remaining eleven hospitals.

**Results** The total response rate was 77% (2113/2760). There were no significant differences in childbirth experience measured with CEQ2 between the groups (induction group, n=354; expectant management group, n=302) in the subscales: *own capacity* (2.8 vs 2.7, p=0.09), *perceived safety* (3.3 vs 3.2, p=0.06) and *professional support* (3.6 vs 3.5, p=0.38) or in the total CEQ2 score (3.3 vs 3.2, p=0.07), respectively. Women in the induction group scored higher in the subscale *participation* (3.6 vs 3.4, p=0.02), although with a small effect size (0.19). No significant difference was observed in overall childbirth experience according to VAS (8.0 (n=735) vs 8.1 (n=735), p=0.22).

**Conclusions** There were no differences in childbirth experience, according to CEQ2 or overall childbirth experience assessed with VAS, between women randomly assigned to induction of labour at 41 weeks or expectant management until 42 weeks. Overall, women rated their childbirth experiences high.

**Trial registration number** ISRCTN26113652.

<div style="border:1px solid #000">

### Strengths and limitations of this study

► The study had a high total response rate.
► The study was based on a large randomised, controlled trial.
► Childbirth experience was evaluated with a validated questionnaire (Childbirth Experience Questionnaire version 2 (CEQ2)) 3 months after birth.
► The questionnaire (CEQ2) was used in only 3 of 14 participating centres; other centres evaluated the overall childbirth experience with a Visual Analogue Scale as an exploratory outcome.
► The response rate with CEQ2 was higher in the induction group than in the expectant management group.

</div>

## INTRODUCTION

Women's childbirth experiences are individual, complex life events, which are related to the outcome for the mother and child.[1–3] These experiences may leave long-lasting impressions, as women tend to remember their childbirth experiences very well.[4–6] A positive birth experience can strengthen the woman and be an empowering life event,[5 7] while a negative experience can increase the risk of postpartum depression,[8] fear of childbirth,[9 10] and future fear of vaginal births.[11] Women with traumatic childbirth experiences have identified the lack and loss of control and insufficient practical and emotional support as the main sources of their traumatic experiences.[12] A qualitative study on women's positive childbirth experiences showed that being in a safe and supporting environment gave them a sense of control and made it possible to focus on strategies to manage childbirth. Knowledge about the birth process was also important, helping the

women to perceive control and enable them to take part in decision-making.[13]

Earlier studies have reported induction of labour being associated with a more negative childbirth experience than spontaneous onset of labour.[3 14 15] However, in prolonged pregnancies, this difference was not as obvious.[3 16] Induction of labour in prolonged pregnancies can relieve some women from feelings of discomfort and the uncertainty of not knowing when and where labour will begin.[17] Women's trust in health professionals, as well as an impatience to deliver a healthy child, can make women to choose and accept induction of labour.[18] Two qualitative interview studies have described that women in a prolonged pregnancy may have growing worries about their unborn child's health and doubts about their body's ability to initiate labour.[19 20] Women also described a lack of information related to late-term pregnancy and not being seen by health professionals.[20]

A systematic review of qualitative studies describes that induction of labour can require a shift in expectations as the woman needs to reconsider her original birth plan to a more medicalised one.[16] To our knowledge, there is one previous randomised, controlled trial (RCT)[21] from Norway comparing women's experiences of induction of labour at 41 weeks with expectant management (n=508). Women in the expectant management group that had not given birth at 42 weeks and 5 days had an induction of labour. The majority of women in both groups stated, 6 months after giving birth, that they would prefer to be induced at 41 weeks instead of expectant management in a future prolonged pregnancy.[21] Another RCT compared childbirth experiences, of nulliparous women aged 35 or older, with induction of labour at 39 weeks with expectant management until spontaneous onset or medical indication for induction of labour, using the Childbirth Experience Questionnaire (CEQ) (n=618). No significant differences were found between the groups.[22]

Induction at or beyond term, compared with expectant management, is related to a lower incidence of perinatal death.[23] In a newly published meta-analysis, it was concluded that induction at 41 weeks compared with expectant management until 42 weeks overall improved perinatal outcome in women without increasing caesarean delivery rate.[24] Still, induction of labour at 42 weeks was the current standard management in Sweden during the time of this study.

The present study is part of the Swedish Post-term Induction Study (SWEPIS), a national, multicentre, randomised, controlled, superiority trial.[25] The aim was to evaluate if induction of labour at 41 weeks (n=1381), compared with expectant management and induction of labour at 42 weeks (n=1379), improved perinatal and maternal outcomes. The study was stopped in advance due to ethical reasons related to six cases of perinatal deaths in the expectant management group and none in the induction group. The rates of perinatal mortality and morbidity (primary composite outcome), caesarean deliveries and instrumental vaginal deliveries were similar in both groups.

A woman's childbirth experience is essential for her psychological well-being and health. Therefore, the aim of this study was to compare childbirth experiences in women randomly assigned to either induction of labour at 41 weeks or to expectant management and induction of labour at 42 weeks.

## METHODS

### Study design

The study reports on childbirth experience within SWEPIS[25 26] and was performed from May 2016 to October 2018 at 14 Swedish hospitals. The trial was a register-based, multicentre, randomised, controlled, superiority trial. It was undertaken with support from the Swedish Network for National Clinical Studies in Obstetrics and Gynecology (www.snaks.se).

### Participants

Eligible participants were women ≥18 years with an uncomplicated singleton pregnancy, with cephalic presentation at 40 weeks+6 days to 41 weeks+1 day. Women were excluded if they had had a previous caesarean delivery or other uterine surgery, pregestational or insulin-dependent gestational diabetes, hypertensive disorder of pregnancy, a multiple pregnancy, breech or transverse position of the fetus, diagnosed oligohydramnios, fetus being small for gestational age or fetal malformation, or contraindications for vaginal delivery.

### Study logistics

The women were given a written leaflet with information about SWEPIS at their regular antenatal check-up at 40 weeks. If interested, they were invited to phone or email the study coordinators, who then gave additional information about the study and booked an appointment at 40 weeks+6 days to 41 weeks+1 day. In the Stockholm region, women were included during a routine ultrasound scan at 41 weeks. At the visit, the study was further explained, eligibility was confirmed and written consent to participate was obtained. Randomisation was done online with a module set up by the Swedish Pregnancy Register.[27] Allocation to the trial group, 1:1, was done using central online randomisation by dynamic allocation, to minimise imbalance between the groups. Study centre and parity (primiparity vs multiparity) were used as minimisation variables. All participants gave written informed consent. Women randomised to the induction group were appointed for induction at 41 weeks+0 days to 41 weeks+2 days. Women randomised to the expectant management group were appointed for induction at 42 weeks+0 days to 42 weeks +1 day if they had not given birth at that time.

### Outcome measures

The Childbirth Experience Questionnaire version 2 (CEQ2) was used as the main outcome measure and was a prespecified secondary outcome in SWEPIS. The CEQ2

was available in Swedish and English. A Visual Analogue Scale (VAS) (1–10), on the woman's overall childbirth experiences, was added as an exploratory outcome in the hospitals where CEQ2 was not distributed.

## Data collection
### Main outcome
Women included at Sahlgrenska University Hospital, Örebro University Hospital, and Falun Hospital were asked to complete the CEQ2 3 months after birth.[28 29] These three hospitals were chosen with the aim of including different sizes of hospitals in Sweden, as well as urban and rural populations. The CEQ2 was sent through a link via email, with two reminders.

### Exploratory outcome
Measurement of overall childbirth experience on a VAS (1–10), within 3 days of delivery, is a clinical routine in Sweden, and was added as an exploratory outcome variable in women giving birth at the remaining 11 centres (Uppsala University Hospital, Södra Älvsborg Hospital, Region Stockholm (n=5 hospitals), Varberg Hospital, Halmstad Hospital, Visby Hospital, and North Älvsborg County Hospital). Data on VAS were retrieved from the Swedish Pregnancy Registry.[27]

Data on background variables, pregnancy and delivery outcomes were also retrieved from the Swedish Pregnancy Register.[27]

### The Childbirth Experience Questionnaire
The first CEQ[30] was developed and validated in Sweden and then translated and culturally adapted to several other languages.[31–34] In the present study, a revised version, CEQ2,[28 29] was used. CEQ2 consists of 22 questions considering four different subscales of the childbirth experience: *own capacity* (eight items), *perceived safety* (six items), *professional support* (five items), and *participation* (three items); see box 1. Responses to the items are given on a 4-point Likert Scale, ranging from 1=totally disagree, 2=mostly disagree, 3=mostly agree and 4=totally agree. Three of the items (perceived pain, control, and sense of security) are rated on a VAS (1–100), which are categorised as 0–40=1, 41–60=2, 61–80=3 and 81–100=4. Negatively worded items are reversed in scoring. Higher scores represent a more positive childbirth experience. The score of each subscale is calculated as a mean of the individual domains. The total CEQ2 score is the mean of the four individual subscale scores (1–4).

### Overall childbirth experience measured with VAS
As part of the clinical routine in Sweden, a question on overall childbirth experience, with a VAS, was used to assess the women's overall childbirth experience within 3 days after delivery. It was added as an exploratory outcome and retrieved for women in the hospitals where CEQ was not distributed. Women were asked by a midwife to rate their childbirth experience on a VAS ranging from 1, representing a very negative experience, to 10, representing a very positive experience.[35] When needed, this could be

---

**Box 1 Subscales and items in the Childbirth Experience Questionnaire version 2**

**Own capacity**
► Labour and birth went as I had expected.
► I felt strong during labour and birth.
► I felt capable during labour and birth.
► I was tired during labour and birth.
► I felt happy during labour and birth.
► I felt that I handled the situation well.
► As a whole, how painful did you feel childbirth was?
► As a whole, how much control did you feel you had during childbirth?

**Perceived safety**
► I felt scared during labour and birth.
► My impression of the team's medical skills made me feel secure.
► I have many positive memories from childbirth.
► I have many negative memories from childbirth.
► Some of my memories from childbirth make me feel depressed.
► As a whole, how secure did you feel during childbirth?

**Professional support**
► Both my partner and I were treated with warmth and respect.
► I would have preferred the midwife to be more present during labour and birth.
► I would have preferred more encouragement from the midwife.
► The midwife conveyed an atmosphere of calm.
► The midwife helped me to find my inner strength.

**Participation**
► I wish the staff had listened to me more during labour and birth.
► I took part in decisions regarding my care and treatment as much as I wanted.
► I received the information I needed during labour and birth.

---

undertaken with the assistance of an interpreter. A rating of 8–10 is considered a very good childbirth experience.[35]

### Data analysis
Mean, SD, median, 25–75 quartiles and 95% CIs were calculated for both CEQ2 and VAS scores. The Mann-Whitney U test was used to compare the differences in childbirth experience (CEQ2 and VAS) between groups and Fisher's exact test for categorical variables. Cohen's effect size was used to estimate clinical differences between groups, where an effect size of 0.2 was considered small, 0.5 medium and 0.8 large.[36] All tests were two-sided, using a significance level of 5%. Results were analysed using SPSS V.25.0 or SAS V.9.

### Patient and public involvement
Participants and public were not involved in the design of the study or in the conduction of the trial. The study findings will be disseminated to the participants and public through popular science articles.

## RESULTS
In all, 2760 participants were included in SWEPIS. Baseline characteristics were similar in the two treatment groups.[25] In total, 68% (656/959) of women responded to CEQ2, 3 months after birth and 81% (1457/1801) of

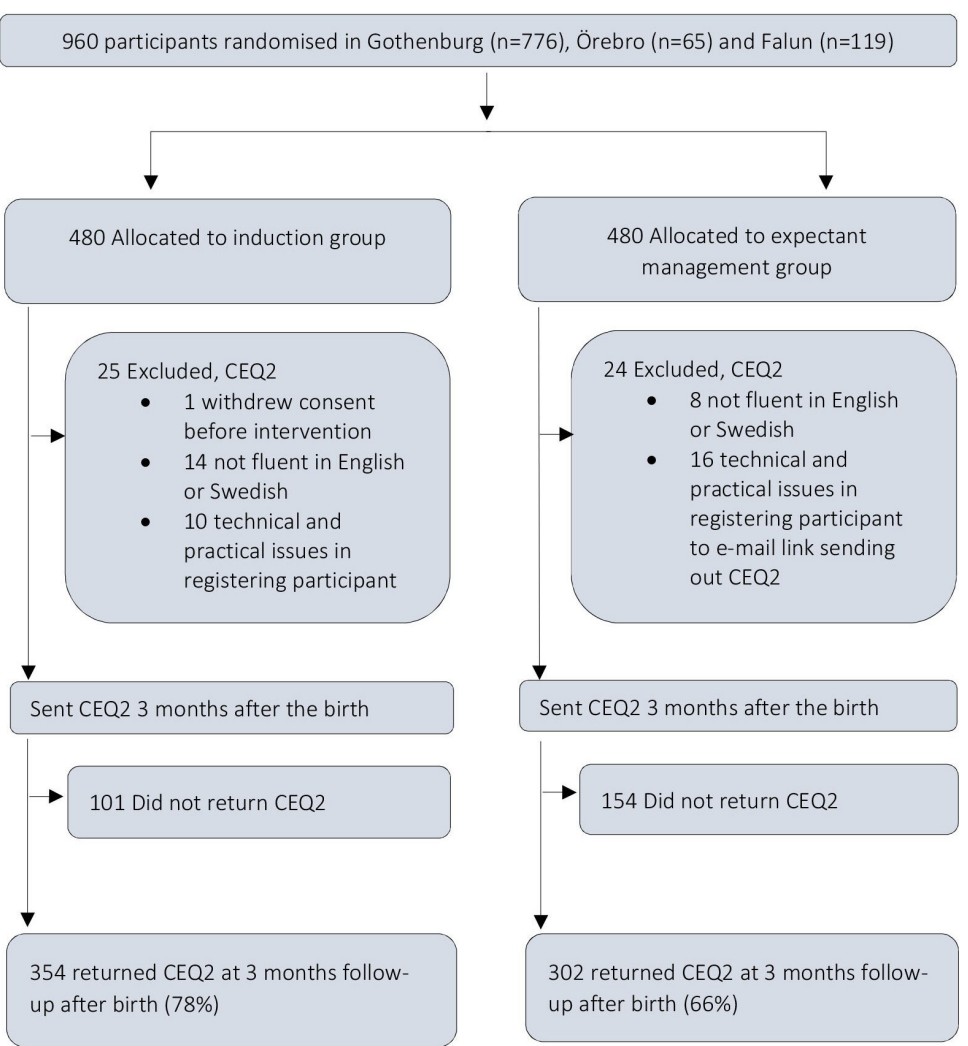

**Figure 1** Flowchart of the women in the two randomised groups that responded to Childbirth Experience Questionnaire version 2 (CEQ2) 3 months after birth.

women rated their overall childbirth experience on a VAS in the remaining centres. In total, 77% responded to either CEQ2 or overall childbirth experience on a VAS.

### Childbirth Experience Questionnaire version 2

In total, 960 women who were randomised at the three centres, Gothenburg (20 May 2016 to 15 October 2018), Falun (23 January 2017 to 15 October 2018), and Örebro (9 September 2017 to 15 October 2018), were asked to fill in the web questionnaire. One woman in the induction group withdrew her consent to participate before induction of labour, and 22 participants were not fluent in Swedish or English and thus did not receive the questionnaire. Due to practical and technical issues, another 26 participants did not receive the questionnaire, most often caused by the woman's email address not being registered at randomisation due to technical issues. A total of 78% of women in the induction group and 66% of women in the expectant management group responded to CEQ2 3 months after giving birth (figure 1).

### Responders and non-responders

Women who responded to and returned the CEQ2 were more likely to have been born in Sweden (88.6%), compared with women who did not respond to CEQ2 (73.4%). More women responding to CEQ2 had a university education (71.6%), compared with the non-responders (59.1%). Of the women responding to CEQ2, 35.4% had spontaneous onset of labour and 67.1% had induction of labour, compared with the non-responders where 47.5% had spontaneous onset of labour and 52.1% had induction of labour (table 1 and online supplemental table A).

### Participant characteristics

For participants responding to CEQ2, the baseline characteristics are presented in table 1. In total, 90.1% of the women in the induction group had their labour induced, compared with 33.4% of the women in the expectant management group. The gestational age at delivery was on average 3 days longer in the expectant management group. The number of newborns admitted to neonatal

**Table 1** Baseline characteristics for participants responding to Childbirth Experience Questionnaire version 2 (CEQ2) and for total population randomised in centres where CEQ2 was sent out.

| Variables | Participants responding to CEQ | | Total CEQ population | |
| --- | --- | --- | --- | --- |
| | Induction group, n=354 | Expectant management group, n=302 | Induction group, n=479 | Expectant management group, n=480 |
| Age at randomisation (years) | | | | |
| Mean (SD) | 31.3 (5.0) | 31.2 (4.2) | 31.1 (4.9) | 31.1 (4.4) |
| Median (IQR) | 31 (28–35) | 31 (28–34) | 31 (28–35) | 31 (28–34) |
| <35 years | 271 (76.6%) | 245 (81.1%) | 370 (77.2%) | 384 (80.0%) |
| ≥35 years | 83 (23.4%) | 57 (18.9%) | 109 (22.8%) | 96 (20.0%) |
| Parity (includes stillborn or live births) | | | | |
| Primiparous | 196 (55.4%) | 164 (54.3%) | 269 (56.2%) | 252 (52.5%) |
| Multiparous | 158 (44.6%) | 138 (45.7%) | 210 (43.8%) | 228 (47.5%) |
| Smoking at first antenatal visit | | | | |
| No | 296/302 (98.0%) | 254/259 (97.1%) | 392/404 (97.0%) | 388/400 (97.0%) |
| Yes | 6/302 (2.0%) | 5/259 (1.7%) | 12/404 (3.0%) | 12/400 (3.0%) |
| BMI at first antenatal visit | | | | |
| Mean (SD) | 25.2 (5.0) | 25.2 (4.8) | 25.3 (5.0) | 25.4 (5.0) |
| Median (IQR) | 24.1 (21.5–27.7), n=302 | 24.2 (21.8–27.6), n=267 | 24.1 (21.5–27.8), n=410 | 24.3 (22.0–27.7), n=417 |
| Region of birth | | | | |
| Sweden | 293/330 (88.8%) | 249/282 (88.3%) | 377/445 (84.7%) | 369/449 (82.2%) |
| Other Nordic countries | 15/330 (4.5%) | 18/282 (6.4%) | 27/445 (6.1%) | 30/449 (6.7%) |
| Europe outside Nordic countries | 6/330 (1.8%) | 4/282 (1.4%) | 7/445 (1.6%) | 6/449 (1.3%) |
| Outside Europe | 16/330 (4.8%) | 14/282 (5.0%) | 34/445 (7.6%) | 44/449 (9.8%) |
| Highest education | | | | |
| Primary school ≤9 years | 9/310 (2.9%) | 5/264 (1.9%) | 14/412 (3.4%) | 11/419 (2.6%) |
| High school 9–12 years | 83/310 (26.8%) | 66/264 (25.0%) | 120/412 (29.1%) | 123/419 (29.4%) |
| University or corresponding | 218/310 (70.3%) | 193/264 (73.1%) | 278/412 (67.5%) | 285/419 (68.0%) |
| Gestational age at delivery (days) | | | | |
| Mean (SD) | 288.0 (1.0) | 291.7 (2.7) | 288.6 (1.2) | 291.5 (2.7) |
| Median (IQR) | 288 (288–289) | 292 (289–294) | 288 (288–289) | 291 (289–294) |
| Time from admittance, to labour ward, to delivery (hours) | | | | |
| Mean (SD) | 20.1 (14.2) | 13.3 (11.0) | 20.3 (14.1) | 13.1 (10.9) |
| Median (IQR) | 15.9 (9.5–27.8) | 10.7 (4.5–19.3) | 16.4 (9.8–27.8) | 10.4 (4.6–19) |
| Onset of birth process | | | | |
| Spontaneous | 31 (8.8%) | 201 (66.6%) | 49 (10.2%) | 153 (68.1%) |
| Scheduled caesarean delivery | 4 (1.1%) | 0 (0.0%) | 4 (0.8%) | 1 (0.2%) |
| Induction | 319 (90.1%) | 101 (33.4%) | 426 (88.9%) | 151 (31.7%) |
| Mode of birth | | | | |
| Spontaneous vaginal | 296 (83.6%) | 256 (84.8%) | 400 (83.5%) | 405 (84.4%) |
| Instrumental vaginal | 24 (6.8%) | 14 (4.6%) | 31 (6.5%) | 26 (5.4%) |
| Caesarean delivery | 34 (9.6%) | 32 (10.6%) | 48 (10.0%) | 49 (10.2%) |
| Use of epidural anaesthesia | 199 (56.2%) | 154 (51%) | 286 (59.7%) | 238 (49.6%) |
| Maternal complications | | | | |
| Perineal lacerations III–IV | 7 (2.0%) | 4 (1.3%) | 11 (2.3%) | 8 (1.7%) |

Continued

| | Participants responding to CEQ | | Total CEQ population | |
|---|---|---|---|---|
| **Variables** | **Induction group, n=354** | **Expectant management group, n=302** | **Induction group, n=479** | **Expectant management group, n=480** |
| Postpartum haemorrhage (>1000 mL) | 38 (10.7%) | 31 (10.3%) | 54 (11.3%) | 45 (9.4%) |
| Postpartum infection | 11 (3.1%) | 6 (2.0%) | 15 (3.1%) | 6 (1.3%) |
| Preeclampsia/gestational hypertension/eclampsia | 2 (0.6%) | 9 (3.0%) | 2 (0.4%) | 14 (2.9%) |
| Perinatal complications | | | | |
| Admittance to neonatal intensive care units | 9 (2.5%) | 20 (6.6%) | 14 (2.9%) | 29 (6.0%) |
| Apgar Score <7 at 5 min | 3/354 (0.8%) | 5/301 (1.7%) | 5/479 (1%) | 5/476 (1.1%) |
| Macrosomia (≥4500 g) | 19 (5.4%) | 30 (9.9%) | 2 (5.0%) | 46 (9.6%) |
| Girls | 165 (46.6%) | 136 (45.0%) | 222 (46.3%) | 213 (44.4%) |
| Birth weight (g) | | | | |
| Mean (SD) | 3811 (430) | 3905 (465) | 3816 (423) | 3897 (459) |
| Median (IQR) | 3800 (3530–4066) | 3898 (3588–4201) | 3800 (3540–4084) | 3900 (3580–4200) |

BMI, body mass index; CEQ, Childbirth Experience Questionnaire.

intensive care units was 2.5% in the induction group and 6.6% in the expectant management group. The rates of spontaneous vaginal births were similar (83.6% vs 84.8% respectively).

### CEQ2 scores
The results of the CEQ2 scores are presented for each domain, as well as the total CEQ score, in table 2. There were no incomplete questionnaires. No significant differences in childbirth experience were identified in three of the four subscales (*own capacity, perceived safety* and *professional support*) or in the total score in the CEQ2 between the two randomised groups. There was a significant difference in the subscale *participation,* where women in the induction group scored slightly higher than women in the expectant management group (p=0.02), with a small effect size (0.19) (figure 2 and table 2).

### Visual Analogue Scale
In total, 1802 women were included at the centres where CEQ2 was not distributed. One woman in the induction group withdrew her consent to participate before induction of labour. A total of 80% of women in the induction group and 82% of women in the expectant management group rated their overall childbirth experience on a VAS (1–10) within 3 days of delivery (online supplemental figure A).

### Responders and non-responders
The women responding to VAS had similar background characteristics to the women who did not answer VAS. Of women responding to VAS, 82.8% were born in Sweden, 61.6% had a university education and 58.7% had their

labour induced versus 79.9%, 62% and 60.5% of the non-responders (online supplemental table B).

### Participant characteristics
In the induction group, 83.9% of women had their labour induced, compared with 33.5% of women in the expectant management group. Women in the expectant management group had a pregnancy, on average 3 days longer, than women in the induction group. There were similar modes of birth in the two groups, 83.2% of women had a spontaneous vaginal birth in the induction group and 82.4% of women in the expectant management group (online supplemental table B).

### VAS score
There was no significant difference, between the induction group and the expectant management group, in overall childbirth experience on VAS or in the number of women rating their childbirth experience as positive (VAS 8–10) (table 3).

### DISCUSSION
The main finding in this study was that there were no significant differences in women's childbirth experiences, between women randomised to induction of labour at 41 weeks and women randomised to expectant management and induction of labour at 42 weeks, in terms of the total CEQ2 score, three of four subscales (*own capacity, perceived safety and professional support*) or according to VAS. However, the women in the induction group scored

**Table 2** Childbirth experience in the induction group and the expectant management group, according to the Childbirth Experience Questionnaire version 2.

| | Induction group, n=354 | Expectant management group, n=302 | Effect size | P value |
|---|---|---|---|---|
| Own capacity | | | 0.16 | 0.09 |
| Mean (SD) | 2.8 (0.6) | 2.7 (0.6) | | |
| Median (IQR) | 2.9 (2.3–3.2) | 2.8 (2.3–3.2) | | |
| (95% CI for mean) | (2.7 to 2.9) | (2.6 to 2.8) | | |
| Perceived safety | | | 0.17 | 0.06 |
| Mean (SD) | 3.3 (0.7) | 3.2 (0.8) | | |
| Median (IQR) | 3.5 (3–3.8) | 3.5 (2.7–3.8) | | |
| (95% CI for mean) | (3.3 to 3.4) | (3.1 to 3.3) | | |
| Professional support | | | 0.09 | 0.38 |
| Mean (SD) | 3.6 (0.5) | 3.5 (0.6) | | |
| Median (IQR) | 3.8 (3.4–4.0) | 3.8 (3.4–4.0) | | |
| (95% CI for mean) | (3.5 to 3.6) | (3.4 to 3.6) | | |
| Participation | | | 0.19 | 0.02 |
| Mean (SD) | 3.6 (0.6) | 3.4 (0.7) | | |
| Median (IQR) | 4.0 (3.3–4.0) | 3.7 (3.0–4.0) | | |
| (95% CI for mean) | (3.5 to 3.6) | (3.4 to 3.5) | | |
| Total CEQ | | | 0.17 | 0.07 |
| Mean (SD) | 3.3 (0.5) | 3.2 (0.6) | | |
| Median (IQR) | 3.5 (3–3.7) | 3.4 (2.9–3.7) | | |
| (95% CI for mean) | (3.3 to 3.4) | (3.2 to 3.3) | | |

P values calculated with Mann-Whitney U test.
CEQ, Childbirth Experience Questionnaire.

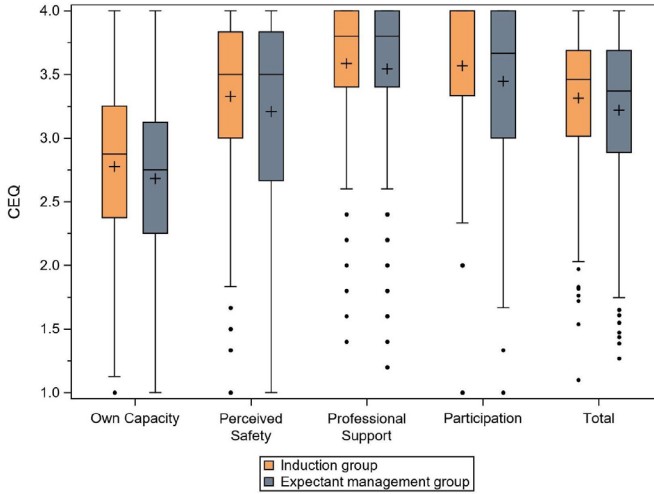

**Figure 2** Childbirth experience in the induction group and the expectant management group, according to the Childbirth Experience Questionnaire (CEQ) version 2. The line in the box refers to the median, and the + refers to the mean.

deliveries or instrumental vaginal births between the two randomised groups,[25] factors that can have a negative impact on women's childbirth experience.[4 30 38]

None of the subscales *own capacity*, *perceived safety*, and *professional support* showed significant differences. Most women have expectations of their capacity to give birth and hope they will be able to use their own inherited physical and psychosocial capacities during labour and birth.[39] It is also reported that most women want a vaginal birth, focusing both on safety and psychosocial well-being.[39] A safe and supporting environment enables the woman to focus on techniques to manage the birth.[13]

Women in the induction group scored significantly higher in the subscale *participation* than the expectant management group (3.6 vs 3.4, maximum score 4, p=0.02). In a systematic review of qualitative studies, Downe *et al*[39] found that most women want a physiological labour and birth. However, if interventions were needed, women wanted sufficient information and to be active in the decision-making to retain a sense of personal achievement and control.[39] This may be reflected in the

slightly higher in the domain *participation*, compared with the women in the expectant management group. Women reported an overall positive childbirth experience.

Induction of labour has been reported to have a negative impact on women's childbirth experience.[15–17] However, this was not observed in our study, even though women had their labour induced to a larger extent in the induction group than in the expectant management group. The women had uncomplicated pregnancies and might have been more motivated for induction of labour, as they were willing to participate in our study. Furthermore, induction can be perceived as a relief from feelings of discomfort in a prolonged pregnancy[17] and a way to meet the child sooner.[37] Another explanation for the results may be that there were no differences in caesarean

**Table 3** Childbirth experience measured with Visual Analogue Scale (VAS) 1–10.

| | Induction group, n=722 | Expectant management group, n=735 | Effect size | P value |
|---|---|---|---|---|
| VAS | | | | |
| Mean (SD) | 8 (1.2) | 8.1 (2) | −0.05 | 0.22 |
| Median (IQR) | 8 (7–10) | 9 (7–10) | | |
| (95% CI for mean) | (7.9 to 8.2) | (8 to 8.3) | | |
| VAS 8–10 | 497 (69%) | 528 (72%) | | 0.23 |

For continuous variables, p values were calculated with the Mann-Whitney U test. For categorical variables, p values were calculated with Pearson $\chi^2$.

domain *participation* as the women who participated in the study may have been more motivated for induction of labour, and therefore, those allocated to the induction group may have experienced being part of the decision-making to a higher degree than those randomised to the expectant management group (standard care).

The mean total CEQ2 score in the women in the induction group was 3.3 and 3.2 in the expectant management group. This is in agreement with earlier studies using the CEQ2. In the RCT by Walker *et al*, women at age 35 or older were randomised to either induction of labour at 39 weeks or expectant management until spontaneous onset or medical indication to induce labour (n=618).[22] The women in the induction group scored a total mean CEQ2 score of 3.03 versus 2.96 (p=0.12) in the expectant management group.[22] Also, women with vaginal delivery scored a mean total CEQ2 score of 3.3 in the CEQ2 validation study.[28]

In a retrospective cohort study from Sweden, including all women with a singleton pregnancy, using the VAS for assessment of overall birth experience, 69% (11 493/16 775) of women scored VAS 8–10 (mean 7.94).[15] These scores are similar to the scores in our study, where 69% in the induction group and 72% in the expectant management group scored 8–10 on VAS.

## Methodological considerations

By using CEQ2[28 30] to compare women's childbirth experiences, we chose an instrument with good psychometric properties.[40] Women could answer the CEQ2 in Swedish or English, which enabled more participants to fill in the questionnaire. The CEQ has been used in several studies in Sweden and has been shown to have good internal and external validity.[41–44] The advantages of CEQ and CEQ2 are that they measure the multidimensional experience of labour and birth. However, there might be dimensions of the childbirth experience that are difficult to capture with a questionnaire.

A larger proportion of women answered the CEQ2 in the induction group (78%) than in the expectant management group (66%). This may be due to women not perceiving they were participating in a study when they were allocated to the expectant management group, which was standard care at the time. Furthermore, women with spontaneous onset of labour did not answer the questionnaire to the same extent as women whose labour was induced. This difference may also be related to that some women having spontaneous onset of labour perceived that they had not participated in the study. However, these differences in response rate between the randomisation groups were not seen for VAS. This might be explained by the fact that the VAS is used as part of the clinical routine including all women in the participating hospitals.

We added the VAS as an exploratory variable with the purpose of obtaining information on overall childbirth experience in the women at the 11 hospitals where the CEQ2 was not distributed. The cause that CEQ2 was not distributed at all participating centres was due to logistic reasons. The advantage of the VAS is that it is part of the

clinical routine and therefore there might be less bias in the women's responses. On the other hand, it is a simplified and non-specific measure, and the women responded to it within 3 days after the birth, compared with the CEQ, which was distributed 3 months after delivery and reflects four domains of childbirth. Hence, the childbirth experience may shift with time when the first wave of relief after the birth has settled. However, the VAS has been shown to correlate to women's overall childbirth experiences and is considered to be useful as a tool to screen for childbirth experience.[15 43 45]

A limitation of the study is that only 22% of eligible women participated in SWEPIS.[25] However, the baseline characteristics were similar for several background variables between the study population in SWEPIS and a Swedish background population of women with uncomplicated pregnancies. However, more women had completed a university education (64% vs 55%) and were born in Sweden (83% vs 73%) in SWEPIS compared with the background population.[25] Another limitation is that oral and written information was given by the midwives at a regular antenatal check-up at 40 weeks, but it is unclear how many women actually received the information. The caregiver's attitudes towards the study, as well as the influence of friends and family, might have affected the women's decision to participate.[46 47] Being part of SWEPIS gave the women a chance to have an induction of labour earlier than in routine care in Sweden; this was also expressed by women at randomisation. It is therefore probable that women participating in SWEPIS were more motivated for or positive to induction of labour than women who declined or chose not to contact the study team. If a specific intervention has negative associations, this may be a reason for not participating and, if it is considered favourable, a reason for participating.[48] Another reason for declining could have been that some women wanted a natural birth without interventions. This reason was expressed by women choosing not to be randomised in an RCT in the Netherlands, which aimed to compare the outcome between induction of labour at 41 weeks and expectant management until 42 weeks. In the same study, women who wanted induction reported lower quality of life (EQ-6D), than women preferring expectant management.[49]

In a qualitative study by Wessberg *et al*,[20] women at 41 weeks expressed lack of information about late-term pregnancy, showing the importance of clear and good information. This was also expressed in a study by Lou *et al*[18] where some women requested more information about their options with late-term pregnancy. Women included in SWEPIS, except in Region Stockholm, received an extra visit to a midwife that explained the procedure of induction of labour and thereby got extra detailed information about possible induction procedures. Therefore, the result of this study should be interpreted from this context.

## Conclusion

The main result of this study was that there were no significant differences in women's childbirth experiences between women randomised to induction of labour at 41 weeks and women randomised to expectant management and induction of labour at 42 weeks. However, women randomised to induction scored higher on the CEQ2 subscale *participation*, but the difference measured with effect size was small. Overall, women's ratings of their childbirth experience were high.

Further research is needed to describe women's experiences of induction of labour in a prolonged pregnancy and how the healthcare system can meet these women's expectations and needs, in order to support them during pregnancy and childbirth.

**Author affiliations**

[1]Institute of Health and Care Sciences, Sahlgrenska Akademy, University of Gothenburg, Gothenbourg, Sweden
[2]Department of Obstetrics, Sahlgrenska University Hospital, Gothenbourg, Sweden
[3]Centre of Perinatal Medicine & Health, Institute of Clinical Sciences, Salgrenska Akademy, Göteborgs Universitet, Gothenbourg, Sweden
[4]Department of Obstetrics and Gynaecology, Faculty of Medicine and Health, Örebro University, Örebro, Sweden
[5]Centre for Clinical Research, Department of Women's Health, Dalarna County Council, Falun, Sweden
[6]Department of Reproductive Medicine, Sahlgrenska University Hospital, Gothenbourg, Sweden
[7]Department of Women's and Children's Health, Uppsala University, Uppsala, Sweden
[8]Department of Obstetrics and Gynaecology, Karolinska University Hospital, Stockholm, Sweden
[9]Department of Women's and Children's Health, Karolinska Institutet, Stockholm, Sweden

**Acknowledgements** The authors wish to thank all the women who participated in the study and took their time to answer the questionnaire, as well as all the midwives and obstetricians at the participating centres. We thank and acknowledge the Swedish Network for National Clinical Studies for their assistance and support.

**Contributors** AW, AD, HH, U-BW, VS and HE conceived and designed the study. HN, AW, HH, U-BW, HF, JW, A-KW and SS oversaw recruitment of study participants and collection of Childbirth Experience Questionnaire data at the local centres. U-BW and HN did the data cleaning together with statisticians Mattias Molin and Per Ekman. HN compiled and analysed the Childbirth Experience Questionnaire version 2 data. Per Ekman did the analysis regarding Visual Analogue Scale. HN wrote the first draft of the manuscript, which was then critically reviewed and revised by all coauthors (HN, AW, AD, HH, U-BW, HF, JW, VS, IL, CB, A-KW, SS and HE). All authors have approved the final version of the manuscript for submission. All authors have full access to all the data in the study and take responsibility for the integrity of the data and the accuracy of the data analysis. HE is the guarantor.

**Funding** This study was supported by the Swedish state under the agreement between the Swedish government and the county councils, the ALF agreement (ALFGBG-440301, ALFGBG-718721, ALFGBG-70940 and ALFGBG-426401); the Health Technology Centre at Sahlgrenska University Hospital; the Foundation of the Health and Medical care committee of the region of Västra Götaland, Sweden (VGFOUREG387351, VGFOUREG640891 and VGFOUREG854081); Hjalmar Svensson Foundation; the foundation Mary von Sydow; born Wijk donation fund; Uppsala-Örebro Regional Research Council (RFR-556711 and RFR-736891); Region Örebro County research committee (OLL-715501); the ALF agreement in Stockholm (ALF-561222,ALF-562222 and ALF-563222); and Centre for Clinical Research Dalarna, Uppsala University, Sweden (CKFUU-417011).

**Disclaimer** The funders had no role in the study design, data collection, data analysis, data interpretation or writing of the report. The researchers were independent of the funders.

**Competing interests** None declared.

**Patient consent for publication** Not required.

**Ethics approval** This study was approved by the Regional Ethics Board in Gothenburg, Sweden, in May 2014 (Dnr: 285–14) and later complementary applications (T 905-15, T 291-16, T 1180-16, T 330-17, T 1066-17, T 087-18, T 347-18, T 961-18 and T 1110-18).

**Provenance and peer review** Not commissioned; externally peer-reviewed.

**Data availability statement** Data are available on reasonable request. Data will be made available, from the corresponding author, on reasonable request.

**ORCID iDs**
Helena Nilvér http://orcid.org/0000-0002-8562-3068
Helena Fadl http://orcid.org/0000-0002-2691-7525
Verena Sengpiel http://orcid.org/0000-0002-3608-7430
Helen Elden http://orcid.org/0000-0003-0000-0476

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
