## [Reviewer comments · BMJ Open]

ARTICLE DETAILS

TITLE (PROVISIONAL)	Women's childbirth experience in the Swedish Post-term Induction Study (SWEPIS): a multicentre, randomised, controlled trial
AUTHORS	Nilvér, Helena; Wessberg, Anna; Dencker, Anna; Hagberg, Henrik; Wennerholm, Ulla-Britt; Fadl, Helena; Wesström, Jan; Sengpiel, Verena; Lundgren, Ingela; Bergh, Christina; Wikström, Anna-Karin; Saltvedt, Sissel; Elden, Helen

VERSION 1 – REVIEW

REVIEWER	Rose Coates City, University of London England
REVIEW RETURNED	14-Sep-2020

GENERAL COMMENTS	Manuscript ID: bmjopen-2020-042340 Women's childbirth experience in the Swedish Post-term Induction Study (SWEPIS): a multicentre, randomised, controlled trial Thank you for the opportunity to review this interesting study of women's childbirth experiences in the SWEPSIS trial, which compared women who had been randomized to induction of labour at 41 weeks gestation, with women randomized to expectant management and induction of labour at 42 weeks. The SWEPSIS trial's primary results are well known and it is positive to see some research coming out of the trial that reports women's experiences of childbirth. The SWEPSIS trial was well-designed and this is reflected in the present manuscript. To be publishable it is my opinion that the current study should focus primarily on the 656 women who completed the Childbirth Experience Questionnaire version 2. To combine these women with a further 1457 who completed a single Visual Analogue Scale (VAS) measuring overall childbirth experience and then conclude that 'There were no differences in childbirth experience according to CEQ2...' does not seem to accurately reflect the CEQ2 results which did show differences, albeit mostly not significant ones, between the groups. The grouping of the 656 CEQ2 women into induction / expectant management groups should also be clearly stated in the abstract. More detailed comments include: Introduction: 1. The Introduction is well-written and references a good combination of qualitative and quantitative literature.2. Page 4, line 47/48. In the sentence 'Women also described a lack of information...' does this relate to a lack of information about induction, or about prolonged pregnancy generally? What is the implication of this?3. Page 4, line 52/53. It would be helpful to state that the only other trial with a similar comparison took place in Norway (reference 20), and to acknowledge (possibly in the Discussion section) that the
---

	health system context may have a role to play in women's decisions about whether they would prefer induction of labour or expectant management. Similarly, reference 21 is a trial that included women aged 35 and older, and it may be expected that this group would prefer induction due to their increased risk of stillbirth. Overall, there needs to be further acknowledgement of the context of induction of labour – the age of the woman, country in which she lives, the method of induction of labour used, and healthcare system variables could all impact decisions about and implications of expectant management or earlier induction of labour. Methods: 4. Page 8, Data Collection. I suggest including the three hospitals that asked women to complete the CEQ2 and not those who completed the VAS. The difference in timepoints and reporting methods of data collection for the two measures (3 months after birth for the CEQ2, self-report) compared with 3 days after birth (single VAS, asked by midwife/healthcare staff) further limits their comparability. 5. Page 8, Outcome Measures section. As the VAS was not a specified outcome of the SWEPSIS trial, it seems more accurate to remove it, and focus on the specified outcome i.e. the CEQ2. 6. It is really useful and transparent to see the items on the CEQ2 included here Results: 7. Page 10, Results. As discussed above, this section would be more accurate if it includes only 656 women who completed the CEQ2. 8. Page 10, lines 25-31. A subheading of responders vs. non-responders (or something similar) would be useful for this paragraph. 9. Page 10, lines 25-31. It may aid the reader to see the demographics of the responders and non-responders written in the following way: Women who completed and returned the CEQ2 were more likely to have been born in Sweden (88.6% of responders vs. 73.4% non-responders), have been university educated (71.6% responders vs. 59.1% non-responders) etc... Clearly there were substantially more women in the induction group who returned the CEQ2 and this needs to be explored thoroughly in the Discussion. 10. Page 12, lines 3-6. A subheading of 'Participant characteristics' or similar would be useful here. 11. Page 12, lines 3-6. A more accurate picture of women's experiences of induced versus spontaneous childbirth would be given if the following four groups were compared on the outcome variables: induction at 41 weeks group with induced labour, induction at 41 weeks group with spontaneous labour, expectant management group with spontaneous labour, expectant management group with induced labour. 12. Were the baseline characteristics of the groups compared statistically? Apologies if I have missed this, but it appears that the following differences need to be discussed in the text, as they are reported in the table: age of women e.g. 23.4% >35 years in induction group vs. 18.9% in expectant management group – why is this? Were women >35 years less likely to take part?; instrumental delivery (6.8% in induction group vs. 4.6% in expectant management group); use of epidural anaesthesia (56.2% induction group vs. 51% expectant management group); preeclampsia etc. (0.6% induction group vs. 3.0% expectant management group). Discussion: 13. The authors are clear on the limitations of the study. It should
--	--

	also be made clear that the findings may be specific to RCTs, i.e. information giving may have been more thorough, and that different study designs are needed to capture the reality of the majority of induction / expectant management of births that take place. 14. It needs to be acknowledged that those who took part would necessarily start at a baseline of not being particularly concerned or have a preference about induction OR about expectant management – and the results reported here may simply reflect that position. Many thanks again for the opportunity to review this manuscript; I hope that the authors find these comments useful in revision of this paper.
--	--

REVIEWER	Diane Quach Monash Health Australia
REVIEW RETURNED	08-Oct-2020

GENERAL COMMENTS	Thank you for the opportunity to review this manuscript, it adds important data around the maternal experience of obstetric interventions to help guide clinical practice and counselling  • Suggest quoting original ethics approval reference number on page 5 under study design • Why did they use different questionnaires at different centres?  o An explanation of why only 3 hospital were chosen to distribute CEQ2 would be helpful o After a few read-throughs it becomes obvious VAS was used as a proxy measure for maternal satisfaction where CEQ2 results were not available, and that VAS is a routinely collected marker of maternal satisfaction within the Swedish health system; hence was chosen that is easily accessible to the researchers. I would suggest making this explanation clearer earlier on so that it is obvious for the first-time reader, most of this information can be moved from the methodological considerations part of the manuscript up to where study design is discussed. • I find it a bit difficult to understand the 3 levels of numbers in Table 3 – could the labelling be fixed?  o If each level corresponds with mean/median/confidence interval – suggest indicating this with symbols • I enjoyed the discussion of methodological considerations. The authors acknowledged and explained pertinent limitations to the study design identified by the reader.
--

REVIEWER	Giuseppe Rizzo Università Roma Tor Vergata Ospedale Cristo Re
REVIEW RETURNED	14-Oct-2020

GENERAL COMMENTS	In this study Authors analyzed women's childbirth experience at 41 weeks of gestation The subject is of interest, the study well designed and written so I would like to congratulate with Authors My comments are  1) I will present the significant data also in a figure as whisker and box plots to give a more direct interpretation of the results 2) I will add in the discussion that the data obtained may be generalized also for induction < 41 weeks a procedure with an increasing incidence
---

REVIEWER	Giuseppe Rizzo
-----------------	----------------

	Università Roma Tor Vergata Ospedale Cristo Re
REVIEW RETURNED	14-Oct-2020

GENERAL COMMENTS	In this study Authors analyzed women's childbirth experience at 41 weeks of gestation The subject is of interest, the study well designed and written so I would like to congratulate with Authors My comments are 1) I will present the significant data also in a figure as whisker and box plots to give a more direct interpretation of the results 2) I will add in the discussion that the data obtained may be generalized also for induction < 41 weeks a procedure with an increasing incidence
--

REVIEWER	Francesco Sera London School of Hygiene & Tropical Medicine London UK
REVIEW RETURNED	08-Nov-2020

GENERAL COMMENTS	This is an interesting study measuring women's childbirth experience within a multicentre RCT. Overall the results suggests no differences either using CEQ questionnaire or VAS. The paper is well structured and clear. The statistical methods are clearly described, and overall the statistical analysis seems coherent with the study design and study objectives. I have only minor concern about the different response rate on CEQ in the induction and the expectant management group. The authors provide information in supplementary table B and in the discussion section (page 15 first paragraph), but I think some more data could help on understanding possible missing mechanism. In particular it would be useful to compare respondent and non-respondent within each treatment group, and it would be useful to compare the two groups also in terms of the VAS. In a sensitivity analysis variables explaining the missing mechanism could be considered as auxiliary variables in a model with CEQ as outcome and treatment groups as exposure. This model would give unbiased estimates under the assumption of missing at random mechanism (given the covariates)
---

VERSION 1 – AUTHOR RESPONSE

Reviewer: 1

Thank you for the opportunity to review this interesting study of women's childbirth experiences in the SWEPSIS trial, which compared women who had been randomized to induction of labour at 41 weeks gestation, with women randomized to expectant management and induction of labour at 42 weeks.

Comments from Reviewer 1	Response
The SWEPSIS trial's primary results are well known and it is positive to see some research coming out of the trial that reports women's experiences of childbirth. The SWEPSIS trial was well-designed and this is reflected in the present manuscript. To be publishable it is my opinion that the current study should focus primarily on the 656 women who completed the	Many thanks for reviewing our manuscript and giving us many useful comments. We agree that CEQ2 is the main focus of this study. However, we still believe that VAS is an important component in the manuscript, as the difference in response-rate that was seen in CEQ2 between the two randomised groups

Childbirth Experience Questionnaire version 2. To combine these women with a further 1457 who completed a single Visual Analogue Scale (VAS) measuring overall childbirth experience and then conclude that ‘There were no differences in childbirth experience according to CEQ2...’ does not seem to accurately reflect the CEQ2 results which did show differences, albeit mostly not significant ones, between the groups. The grouping of the 656 CEQ2 women into induction / expectant management groups should also be clearly stated in the abstract.	were not found in the single VAS question. To make it more clear, we have now adjusted the text so that VAS is presented as an exploratory outcome and it is now clear that VAS is retrieved as part of the clinical routine. We did conclude ‘There were no differences in childbirth experience..’ in spite of that the score in the subscale “Participation” was slightly higher in the induction group as the effect size was small and the total CEQ score (and VAS) did not differ between groups.
--	--

Introduction:

Comments from Reviewer 1	Response
1. The Introduction is well-written and references a good combination of qualitative and quantitative literature.	Thank you.
2. Page 4, line 47/48. In the sentence ‘Women also described a lack of information...’ does this relate to a lack of information about induction, or about prolonged pregnancy generally? What is the implication of this?	We have now clarified that the lack of information described by the women were related to prolonged pregnancy.
3. Page 4, line 52/53. It would be helpful to state that the only other trial with a similar comparison took place in Norway (reference 20), and to acknowledge (possibly in the Discussion section) that the health system context may have a role to play in women’s decisions about whether they would prefer induction of labour or expectant management. Similarly, reference 21 is a trial that included women aged 35 and older, and it may be expected that this group would prefer induction due to their increased risk of stillbirth. Overall, there needs to be further acknowledgement of the context of induction of labour – the age of the woman, country in which she lives, the method of induction of labour used, and healthcare system variables could all impact decisions about and implications of expectant management or earlier induction of labour.	We have clarified that the trial took place in Norway and that in this trial expectant management was allowed until week 42+5 . For reference 21, we have rephrased the sentence so that it becomes clearer that this study included women aged 35 or older. Furthermore, we have added some clarifications regarding women’s preferences for induction of labour versus expectant management both in the background section as well as in the last paragraph of the Methodological considerations in the Discussion section.

Methods:

Comments from Reviewer 1	Response
4. Page 8, Data Collection. I suggest including the three hospitals that asked women to complete the CEQ2 and not those who completed the VAS. The difference in timepoints and reporting methods of data collection for the two measures (3 months after birth for the CEQ2, self-report) compared with 3 days after birth (single VAS, asked by midwife/healthcare staff) further limits their comparability.	We agree that these two measures are not comparable due to differences in how they were collected. We have now presented them as main outcome and exploratory outcome, to further enlighten that they are different but complementary measures.

5. Page 8, Outcome Measures section. As the VAS was not a specified outcome of the SWEPSIS trial, it seems more accurate to remove it, and focus on the specified outcome i.e. the CEQ2.	We believe that the VAS score still can add some valuable information even though it is a rather rough overall measure which does not offer detailed information. It is more and more common to use VAS in Sweden to retrieve information on a group level about women's experiences. So we think it can be of value as a complementary outcome. Although, we have now adjusted the text so that VAS is presented as an exploratory outcome and so that it is more clear that it is retrieved as part of the clinical routine.
6. It is really useful and transparent to see the items on the CEQ2 included here	Thank you for your comment.

Results:

Comments from Reviewer 1	Response
7. Page 10, Results. As discussed above, this section would be more accurate if it includes only 656 women who completed the CEQ2.	We believe that the VAS score still can add some valuable information (please see above).
8. Page 10, lines 25-31. A subheading of responders vs. non-responders (or something similar) would be useful for this paragraph.	Thank you for this suggestion. We have now added subheading for this section as well as for most paragraphs in the methods to make this section more accessible.
9. Page 10, lines 25-31. It may aid the reader to see the demographics of the responders and non-responders written in the following way: Women who completed and returned the CEQ2 were more likely to have been born in Sweden (88.6% of responders vs. 73.4% non-responders), have been university educated (71.6% responders vs. 59.1% non-responders) etc... Clearly there were substantially more women in the induction group who returned the CEQ2 and this needs to be explored thoroughly in the Discussion.	Thank you for this suggestion. We have now changed the phrasing accordingly.
10. Page 12, lines 3-6. A subheading of 'Participant characteristics' or similar would be useful here.	Thank you for this suggestion. We have now added subheading for this section as well as for most paragraphs in the methods section to make it easier to read.
11. Page 12, lines 3-6. A more accurate picture of women's experiences of induced versus spontaneous childbirth would be given if the following four groups were compared on the outcome variables: induction at 41 weeks group with induced labour, induction at 41 weeks group with spontaneous labour, expectant management group with spontaneous labour, expectant management group with induced labour.	We understand that such a comparison could be of interest in retrospect. However, in the present study the aim was to compare women experiences in the two pre-designated randomized groups.
12. Were the baseline characteristics of the groups compared statistically? Apologies if I have missed this, but it appears that the following differences need to be discussed in the text, as they are reported in the table: age of	We have now included the total population that was randomised at the three centres where CEQ was assessed in Table 2. The ages were comparable between the randomised groups.

women e.g. 23.4% >35 years in induction group vs. 18.9% in expectant management group – why is this? Were women >35 years less likely to take part?; instrumental delivery (6.8% in induction group vs. 4.6% in expectant management group); use of epidural anaesthesia (56.2% induction group vs. 51% expectant management group); preeclampsia etc. (0.6% induction group vs. 3.0% expectant management group).	The differences in the outcomes of SWEPIIS (instrumental delivery, use of epidural analgesia and preeclampsia) are discussed in the main SWEPIIS article (Wennerholm et al, 2019). Here we have included it as background information, as the main focus of this paper is to present women’s childbirth experiences.
--	--

Discussion:

Comments from Reviewer 1	Response
13. The authors are clear on the limitations of the study. It should also be made clear that the findings may be specific to RCTs, i.e. information giving may have been more thorough, and that different study designs are needed to capture the reality of the majority of induction / expectant management of births that take place.	Thank you for this comment. We have clarified this in the end of the discussion section.
14. It needs to be acknowledged that those who took part would necessarily start at a baseline of not being particularly concerned or have a preference about induction OR about expectant management – and the results reported here may simply reflect that position.	We agree that the women’s baseline view and experience may well influence on the results to some extent. We do believe that women participating in this study were motivated to have an induction. We have further clarified this in the last paragraph of the Methodological consideration section. We added a new reference, Keulen et al., 2020, that did look at reasons for women preferring either induction of labour at 41 gestational weeks to expectant management at 42 weeks, as the results may also be relevant to our study.
Many thanks again for the opportunity to review this manuscript; I hope that the authors find these comments useful in revision of this paper.	Thank you again for reviewing our manuscript and giving many helpful comments.

Reviewer: 2

Thank you for the opportunity to review this manuscript, it adds important data around the maternal experience of obstetric interventions to help guide clinical practice and counselling

Comments from Reviewer 2	Response
Suggest quoting original ethics approval reference number on page 5 under study design	The ethical approval is included at the end of the article under Ethics approval in accordance to authors’ instructions.
Why did they use different questionnaires at different centres? o An explanation of why only 3 hospital were chosen to distribute CEQ2 would be helpful	CEQ2 was not distributed at all participating centres in SWEPIIS due to logistic reasons. We have now clarified this in the methodological discussion section, third paragraph.
o After a few read-throughs it becomes obvious VAS was used as a proxy measure for maternal satisfaction where CEQ2 results were not available, and that VAS is a routinely collected marker of maternal satisfaction within	We have revised the section with VAS so that it earlier becomes more clear that VAS is an exploratory outcome and retrieved as part of the clinical routine.

the Swedish health system; hence was chosen that is easily accessible to the researchers. I would suggest making this explanation clearer earlier on so that it is obvious for the first-time reader, most of this information can be moved from the methodological considerations part of the manuscript up to where study design is discussed.	
I find it a bit difficult to understand the 3 levels of numbers in Table 3 – could the labelling be fixed? o If each level corresponds with mean/median/confidence interval – suggest indicating this with symbols	Thank you for this suggestion. We have now clarified the tables accordingly both in the manuscript and the supplementary files.
I enjoyed the discussion of methodological considerations. The authors acknowledged and explained pertinent limitations to the study design identified by the reader.	Thank you very much for reviewing our manuscripts and for your helpful comments.

Reviewer: 3

In this study Authors analyzed women's childbirth experience at 41 weeks of gestation The subject is of interest, the study well designed and written so I would like to congratulate with Authors

Comments from Reviewer 3	Response
1) I will present the significant data also in a figure as whisker and box plots to give a more direct interpretation of the results	Thank you for this excellent suggestion. We have now added a box-plot as Figure 2.
2) I will add in the discussion that the data obtained may be generalized also for induction < 41 weeks a procedure with an increasing incidence	Thank you for your comment. We would prefer to be more cautious when interpreting the results. To make it more clear, we have added a reference (Keulen et al, 2020) in the last paragraph in the section about methodological consideration. It is a reference from a study similar to ours, where women were asked to fill in a questionnaire about their reasons for wanting or not wanting induction of labour at gestational week 41 or expectant management. Thank you very much for reviewing our manuscript and for your helpful comments.

Reviewer: 4

Comments to the Author

Comments from Reviewer 4	Response
This is an interesting study measuring women's childbirth experience within a multicentre RCT. Overall the results suggests no differences either using CEQ questionnaire or VAS. The paper is well structured and clear. The statistical methods are clearly described, and overall the statistical analysis seems coherent with the study design and study	Thank you so much for reviewing our manuscript and for your helpful comments.

objectives.	
I have only minor concern about the different response rate on CEQ in the induction and the expectant management group. The authors provide information in supplementary table B and in the discussion section (page 15 first paragraph), but I think some more data could help on understanding possible missing mechanism. In particular it would be useful to compare respondent and non-respondent within each treatment group, and it would be useful to compare the two groups also in terms of the VAS. In a sensitivity analysis variables explaining the missing mechanism could be considered as auxiliary variables in a model with CEQ as outcome and treatment groups as exposure. This model would give unbiased estimates under the assumption of missing at random mechanism (given the covariates)	Thank you for this comment. This is a limitation of this study. We understand this concern and agree with the reviewer. In Table 2 we have added further information about the total population randomised in the three centres where CEQ2 was assessed. Like this, it becomes easier for the reader to compare the respondents of CEQ2 to the women randomised. In Supplementary Table B we have added this information for participants in the remaining centres answering to VAS and also included the information that was earlier available in Supplementary Table C. This issue is further clarified in the discussion section.

VERSION 2 – REVIEW

REVIEWER	Diane Quach Monash Health / Monash University Australia
REVIEW RETURNED	02-Jan-2021

GENERAL COMMENTS	Thank you for the opportunity to review the revised version of this manuscript. I believe it largely is ready for publication and only have some small suggestions as below:  - Suggest moving paragraphs under sub-headings 'main outcome' and 'exploratory outcome' to sit under heading 'Outcome measures' - Page 10 line 29 'withdraw' should be 'withdrew' - Page 10 line 40 sentence beginning 'As well did more women ... ' can be reworded to be clearer and grammatically correct
--

REVIEWER	Francesco Sera London School of Hygiene and Tropical Medicine, London, UK
REVIEW RETURNED	11-Jan-2021

GENERAL COMMENTS	The authors have answered positively to my comments. I have no further comments
---

VERSION 2 – AUTHOR RESPONSE

Answers to reviewers:

Many thanks for reviewing the revised version of our manuscript and for giving us useful comments.

Comments from Reviewer 2:

- Suggest moving paragraphs under sub-headings 'main outcome' and 'exploratory outcome' to sit under heading 'Outcome measures'

Answer: We have changed the order of the headings so that the header 'Outcome measures' are first and the header 'Data collection' with the sub-headings 'main outcome' and 'exploratory outcome' comes second. Thank you for the suggestion, it makes more sense like this.

- Page 10 line 29 'withdraw' should be 'withdrew'

Answer: Thank you for noticing this mistake. We have now corrected it.

- Page 10 line 40 sentence beginning 'As well did more women ... ' can be reworded to be clearer and grammatically correct

Answer: We have now reworded the sentence so that it is clearer.